

# Integrated network pharmacology and molecular docking approaches to reveal the synergistic mechanism of multiple components in *Venenum Bufonis* for ameliorating heart failure

Wei Ren[1,*], Zhiqiang Luo[2,*], Fulu Pan[3,*], Jiali Liu[1], Qin Sun[1], Gang Luo[1], Raoqiong Wang[1], Haiyu Zhao[4], Baolin Bian[4], Xiao Xiao[5], Qingrong Pu[1], Sijin Yang[1] and Guohua Yu[2]

[1] National Traditional Chinese Medicine Clinical Research Base, Affiliated Traditional Chinese Medicine Hospital, Southwest Medical University, Luzhou, China
[2] School of Life Sciences, Beijing University of Chinese Medicine, Beijing, China
[3] School of Chinese Materia Medica, Beijing University of Chinese Medicine, Beijing, China
[4] Institute of Chinese Materia Medica, China Academy of Chinese Medical Sciences, Beijing, China
[5] Beijing National Laboratory for Molecular Sciences, Key Laboratory of Analytical Chemistry for Living Biosystems, CAS Research/Education Center for Excellence in Molecular Sciences, Institute of Chemistry, Chinese Academy of Sciences, Beijing, China
[*] These authors contributed equally to this work.

Corresponding authors
Sijin Yang, ysjimn@sina.com
Guohua Yu, 202002016@bucm.edu.cn

## ABSTRACT

*Venenum Bufonis* (VB), also called Chan Su in China, has been extensively used as a traditional Chinese medicine (TCM) for treating heart failure (HF) since ancient time. However, the active components and the potential anti-HF mechanism of VB remain unclear. In the current study, the major absorbed components and metabolites of VB after oral administration in rats were first collected from literatures. A total of 17 prototypes and 25 metabolites were gathered. Next, a feasible network-based pharmacological approach was developed and employed to explore the therapeutic mechanism of VB on HF based on the collected constituents. In total, 158 main targets were screened out and considered as effective players in ameliorating HF. Then, the VB components–main HF putative targets–main pathways network was established, clarifying the underlying biological process of VB on HF. More importantly, the main hubs were found to be highly enriched in adrenergic signalling in cardio-myocytes. After verified by molecular docking studies, four key targets (ATP1A1, GNAS, MAPK1 and PRKCA) and three potential active leading compounds (bufotalin, cinobufaginol and 19-oxo-bufalin) were identified, which may play critical roles in cardiac muscle contraction. This study demonstrated that the integrated strategy based on network pharmacology and molecular docking was helpful to uncover the synergistic mechanism of multiple constituents in TCM.

## INTRODUCTION

Heart failure (HF), a multifactorial degenerative disease, occurs when the heart is not able to pump blood efficiently to satisfy the oxygen and nutritional needs of the body (*Asano et al., 2019*). HF affects over 26 million people worldwide and continues to represent a major burden for public health due to its high mortality, morbidity and healthcare expenses (*Di Palo & Barone, 2020*; *Lother & Hein, 2016*). The incidence rate of HF rises in magnitude with age and the major etiologies were coronary heart disease, abnormal heart valves and hypertension (*Akkineni et al., 2019*). Western medicines, such as angiotensin converting enzyme (ACE) inhibitors, diuretics, $\beta$-adrenergic blockers, angiotensin receptor I antagonists and positive inotropic agents, are currently the main treatment programs for HF (*Xu et al., 2020*; *Yang et al., 2020a*; *Yang et al., 2020b*). However, long-term use of these chemical drugs will lead to a series of adverse reactions like electrolyte depletion, fluid depletion, and hypotension (*Jia et al., 2020*). Therefore, novel alternative or synergetic anti-HF therapies are greatly needed.

*Venenum Bufonis* (VB) is the dried white secretion of the auricular and skin glands of *Bufo bufo gargarizans* Cantor or *Bufo melanostictus* Schneider (*He et al., 2019*; *Yun et al., 2009*). As a precious traditional Chinese medicine (TCM), VB has long been used for treating heart failure, arrhythmia, swells, sore throat, pains, cancers and many other diseases (*Liang et al., 2008*; *Pan et al., 2020*). Extensive natural product studies have indicated that VB contains a high level of bufadienolides and many other constituents like alkaloids, cyclic amides, sterols, polypeptides, proteins, polysaccharides, and organic acids (*Wei et al., 2019*). Modern pharmacological studies have confirmed that VB exhibited a variety of pharmacological effects, including cardiotonic, antinociceptive, anti-tumor, anesthetic, anti-inflammatory as well as antimicrobial properties (*Wei et al., 2020*). Importantly, it is evident that VB exerts a strong cardiac excitatory effect like that of digitalis, and the drug possesses many advantages such as no accumulation, quick-acting and diuretic action (*Wei et al., 2019*). According to the Chinese Pharmacopoeia (2015 edition), VB is contained in many TCM prescriptions for the treatment of HD, such as Jiuxin Pill and Shexiang Baoxin Pill (*Chinese Pharmacopoeia Commission, 2015*). Although well-practiced in clinical medicine, the holistic pharmacological mechanisms of VB on HF are largely unknown.

As an emerging field of pharmacology, network pharmacology delivers a systematic and holistic understanding of drug action and disease complexity, which shares key ideas with the integrality and systematicness of TCM theory (*Luo et al., 2020*; *Zhang et al., 2020*). An increasing body of evidence suggests that network pharmacology is a powerful tool to illuminate the integration synergistic mechanism of action of TCM from the multi-dimensional perspective (*Chen et al., 2019*; *Miao et al., 2019*). For example, the bioactive candidates and underlying mechanisms of *Cichorium glandulosum* for ameliorating type 2 diabetes mellitus was successfully elucidated using "compound–target" network analysis (*Qin et al., 2019*). The action mechanism of *Carthamus tinctorius* L. on cardiovascular disease was also elaborated based on the "compound-protein/gene-disease" network (*Yu et al., 2019*). However, due to the limitation of this method, many previous researches usually collected TCM components from related TCM databases to establish the compound–target

map. The candidate compounds might not be in line with the components actually delivered into the blood circulatory system, which unavoidably produced unreal results (*Ding et al., 2019*; *Zhang et al., 2018*).

In this study, first the in vivo ingredients of VB after oral administration in rats were taken from the literature. Second, the VB- and HF-associated targets were predicted. Third, network construction and pathway enrichment analysis were used to explore the active components and the potential targets relevant to the treatment of HF with VB. Finally, molecular docking was performed to confirm the specific interactions between VB and the candidate targets. The above study not only provides a comprehensive understanding about the molecular mechanism of VB acting on HF, but also offers a rapid and effective strategy for screening desired compounds from TCM. The flowchart was illustrated in Fig. 1.

# MATERIALS AND METHODS

## In vivo constituents of VB
A total of 42 in vivo constituents of VB have been reported, including 17 prototypes and 25 metabolites (*He et al., 2012*; *Liang et al., 2008*; *Miyashiro, Nishio & Shimada, 2008*; *Ning et al., 2010*; *Tao et al., 2017*; *Xia et al., 2010*; *Xin et al., 2016*; *Zhu et al., 2013*). The molecular 2D files of these constituents were downloaded from the ChemSpider database (http://www.chemspider.com/) and were saved in mol format (step 1 in Fig. S1). The details were listed in Table S1.

## The prediction of VB-related targets
The predicted proteins targeted by the in vivo constituents of VB were screened from MedChem Studio (MedChem Studio, 3.0; Simulations Plus, Inc, Lancaster, CA, USA, 2012). This software could efficiently capture the FDA-approved drugs that have similar chemical structures to the components in TCM (*Yu et al., 2016*). We picked out VB compound-drug pairs with high confidence scores ($\geq 0.6$) and considered the target proteins of the known drugs as the VB-related targets. Other parameters were set as the default values (step 1 in Fig. S1).

## Known therapeutic targets for HF
In this study, two databases were employed in acquiring pathological targets for HF, by use of "heart failure" as the query. One was the DrugBank database (http://www.drugbank.ca/, version 5.1.1), which could provide detailed information on the drug targets and their links with human diseases (*Griesenauer, Schillebeeckx & Kinch, 2019*). Only drug–target interactions for FDA-approved anti-HF drugs and potential human protein targets associated with HF were selected for further analysis. The second platform was the Online Mendelian Inheritance in Man (OMIM) database (http://www.omim.org/, updated on May 4, 2018), a constantly updated database of human genetic diseases and genes (step 1 in Fig. S1) (*Hamosh et al., 2005*).

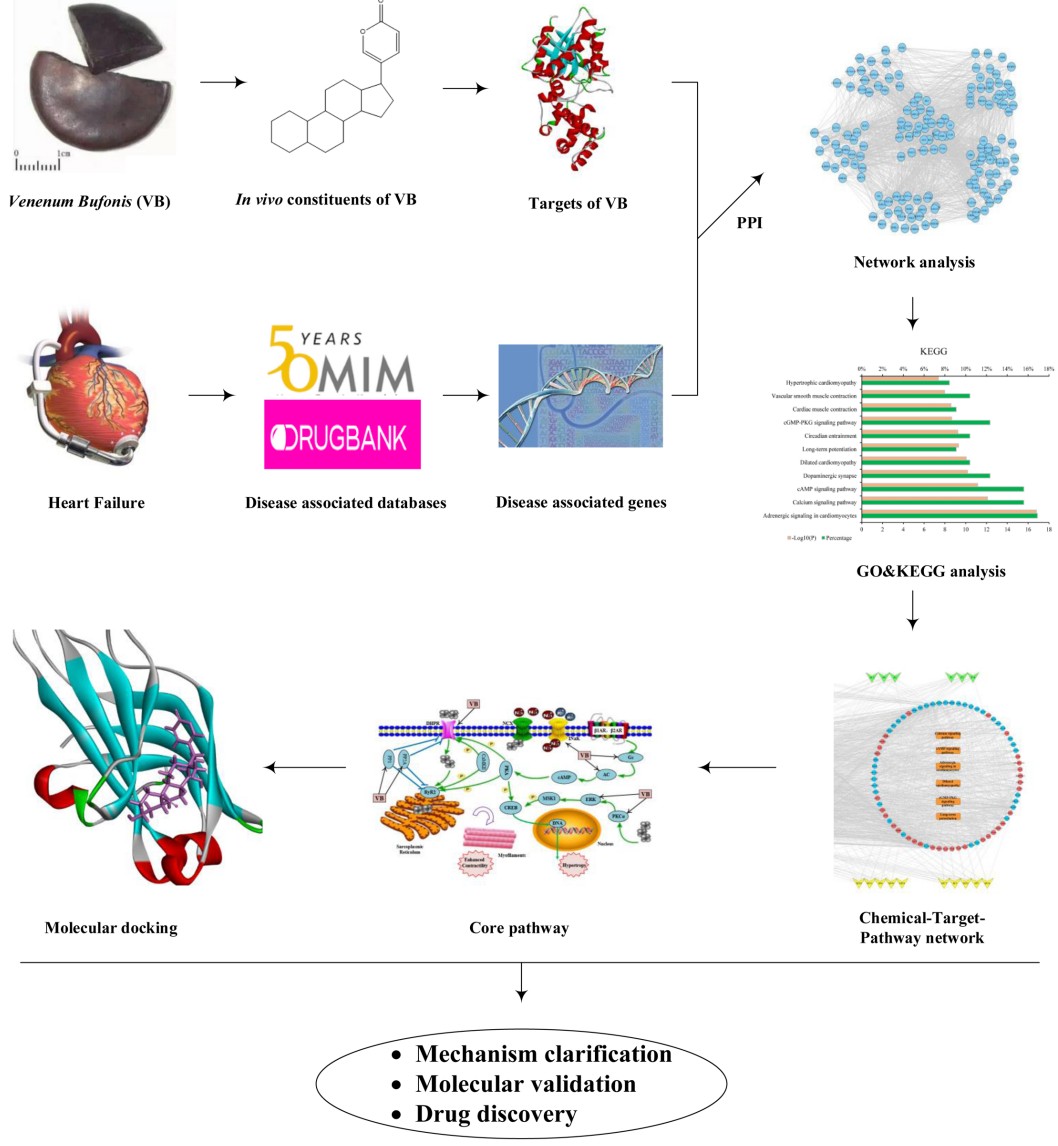

**Figure 1** **Workflow of network pharmacology and molecular docking approaches to reveal the active components and molecular mechanisms of VB acting on HF.**

## Protein–protein interaction (PPI) data

The PPI information related to the putative targets of VB constituents and known therapeutic targets for HF was harvested by use of STRING (Search Tool for the Retrieval of Interacting Genes/Proteins) database (http://string-db.org/). This database could provide a global perspective of proteins and their functional interactions and associations (*Jensen et al., 2009*). Results were limited to "*Homo sapiens*" and protein interactions with a confidence score greater than or equal to 0.4 would be selected. Other parameters were set as the default values (step 2 in Fig. S1).

## Network construction and analysis

In order to illustrate the interaction among the ingredients, targets and diseases, a "ingredient–target–disease" network was built by introducing the information of candidate compounds of VB, putative targets of VB and HF-associated targets into Cytoscape software (version 3.6.0, Boston, MA, USA). This software is an efficient open source bioinformatics tool for visualizing and analysing the complex biological networks (*Shannon et al., 2003*). Three important topological characteristics ("degree", "betweenness", and "closeness"), which have been described in our previous publication (*Yu et al., 2018*), were calculated to assess the central attribute of each hub node in the network by use of the Cytoscape plugin "Network Analyzer". And "degree" > median degree centrality, "betweenness" > median betweenness centrality and "closeness" > median closeness centrality were adopted as the screening criteria to acquire the critical targets (*Luo et al., 2020*). Other parameters were set as the default values (step 2 in Fig. S1).

## Pathway enrichment analysis

Kyoto Encyclopedia of Genes and Genomes (KEGG) pathways analysis and Gene Ontology (GO) enrichment analysis were undertaken to explore the potential functions of the pivotal target proteins involved in the VB-mediated treatment of Hf (*Nguyen et al., 2019a*; *Nguyen et al., 2019b*), by use of the Database for Annotation, Visualization and Integrated Discovery (DAVID) system (http://david.abcc.ncifcrf.gov/home.jsp/, v6.8) (*Dennis Jr et al., 2003*). Relevant pathways with the false discovery rate (FDR)-corrected *P*-value < 0.05 were considered statistically significant. Other parameters were set as the default values (step 3 in Fig. S1).

## Molecular docking simulation

Molecular docking studies were conducted to further verify the reliability of the potential targets using CDOCKER module implemented in Discovery Studio 2016 (DS 2016). CDOCKER is a semi-flexible molecular docking analysis method based on the CHARMm force field, which can produce high-precision docking results, and it provides information on the interaction binding energy and ligand–receptor docking mode. The three-dimensional (3D) structures of the candidate compounds were generated using Chem3D Pro 12.0 and the crystallographic structures of the proteins encoded by the candidate target genes were obtained from the PDB database (http://www.rcsb.org/pdb/home/home.do), which was then decorated by removing the ligands, adding hydrogen, removing water, optimizing and patching amino acids. The binding site was defined by the ligand atoms, and the radius range was automatically generated. After each compound was docked, the 10 best conformations were obtained (Yang, 2020). Finally, CDOCKER interaction energies (CIEs) were used to assess the binding affinities between the core targets and the corresponding compounds. And the conformation corresponding to the lowest CIE was selected as the most probable binding conformation. All parameters used in calculation were default except for explained (step 4 in Fig. S1).

## RESULTS AND DISCUSSION

### Putative targets for VB

As shown in Table S2, a total of 260 potential targets of the in vivo components of VB were obtained by MedChem Studio. The results showed that the candidate compounds could act on multiple targets, and one target could also be linked to multiple components.

### Known therapeutic targets for HF

We collected 109 and 199 HF-related targets from DrugBank database and OMIM database, respectively. We then checked the data and eliminate redundant entries, leaving a final dataset of 292 targets associated with HF (Table S3). Among them, 22 targets were both the VB- and HF-related targets, including NR1I2, MT-CO1, CA2, NR3C1, SHBG, ATP1A1, CYP17A1, PGR, NR3C2, MME, AR, ACE, PRKCA, CYP11B2, RXRA, PPARG, PIK3CG, PPARD, PDE4B, KCNMA1, MED1 and HIF1A.

### Network and pathway analysis

To facilitate scientific interpretation of the complex relationships between VB and HF, a "chemical target-disease associated gene" network, comprising the VB-related targets and the HF-related targets, was constructed based on the PPI data from STRING database. As listed in Table S4, this network was composed of 461 nodes and 5009 edges. After computing the values of the topological features of all hubs, 158 major hubs were identified because they satisfied the criteria (degree cutoff = 19, betweenness cutoff = 0.001659, and closeness cutoff = 0.381426). Among them, 93 hubs were VB-related targets, 65 hubs were HF-related targets. These major hubs may play a critical role in the entire interaction network. The specific information about the major hubs was shown in Table S5.

### Potential mechanisms of VB in treating HF

To elucidate the biological process (BP), molecular function (MF), cellular components (CC) of these major hubs involved in, GO enrichment analysis was conducted on the major hubs. As shown in Figs. 2A–2C, the top 10 significant GO entries ($P < 0.05$) were selected on the basis of $P$ value.

To deeply determine the function and systemic association of the main hubs, KEGG pathway enrichment analysis were conducted. The results indicated that these hubs were enriched in 101 significant pathways ($P < 0.05$). As shown in Fig. 2D, the top 11 pathways were considered as the main biological processes involved in the treatment. To further identify the functional mechanisms of VB on HF, the candidate compounds-main VB-related targets-main pathways network diagram was generated and elucidated in Fig. 3. The hubs can be mainly divided into the following three functional modules (circles): signal transduction (including adrenergic signalling in cardio-myocytes, calcium signalling pathway, cAMP signalling pathway, dilated cardiomyopathy, long-term potentiation and cGMP-PKG signalling pathway), cardiovascular system (including cardiac muscle contraction, vascular smooth muscle contraction and hypertrophic cardiomyopathy), and neural regulation (including dopaminergic synapse and circadian entrainment). Interestingly, adrenergic signalling in cardio-myocytes was highly enriched in KEGG

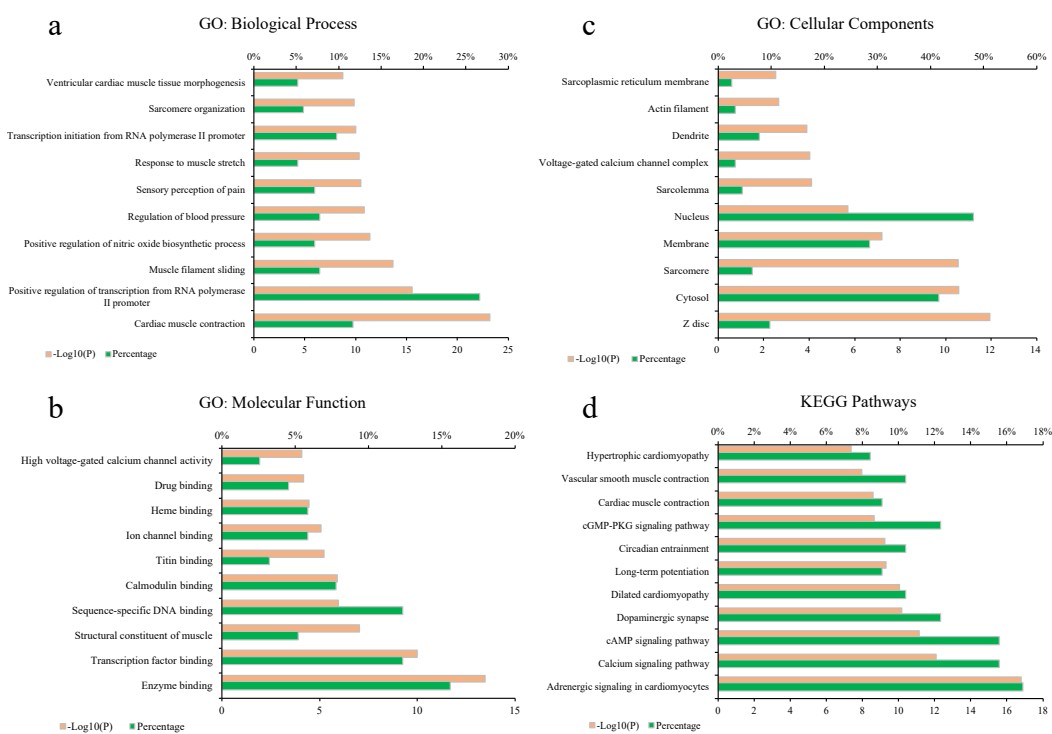

**Figure 2  GO term performance and pathway enrichment analysis of the major hubs.** (A) GO: BP; (B) GO: MF; (C) GO: CC; and (D) KEGG. The ordinate stands for GO terms or the main pathways, the primary abscissa stands for minus log $10(P)$, and the secondary abscissa stands for the percentage of major hubs involved in the corresponding GO terms or the main pathways out of total major hubs.

pathway analysis, which played a critical role in the regulation of cardiac muscle contraction (the top-ranked GO: Biological Process terms), suggesting that VB may impart therapeutic effects on HF majorly through adrenergic signalling in cardio-myocytes.

As shown in Table S6, the VB putative targets associated with adrenergic signalling in cardio-myocytes include guanine nucleotide-binding protein G(s) subunit alpha (GNAS), adenylate cyclase 2 (ADCY2), ADCY5, protein phosphatase 1 catalytic subunit gamma (PPP1CC), protein phosphatase 2 catalytic subunit alpha (PPP2CA), protein phosphatase 2 catalytic subunit beta (PPP2CB), calcium voltage-gated channel subunit alpha1 C (CACNA1C), CACNA1D, ATPase Na$^+$/K$^+$ transporting subunit alpha 1 (ATP1A1), mitogen-activated protein kinase 1 (MAPK1), protein kinase C alpha (PKC$\alpha$, encoded by PRKCA). Figure 4 depicts a graphical overview of adrenergic signalling in cardio-myocytes influenced by main putative targets of VB components.

## Molecular docking

Table 1 displayed the CIEs (top 5 for each target) of VB hit constituents against the active sites of the screened targets in adrenergic signalling, including ATP1A1, GNAS, ADCY2, ADCY5, PPP1CC, PPP2CA, PPP2CB, CACNA1C, CACNA1D, MAPK1 and PRKCA. The results indicated that the VB-related components had been docked successfully with

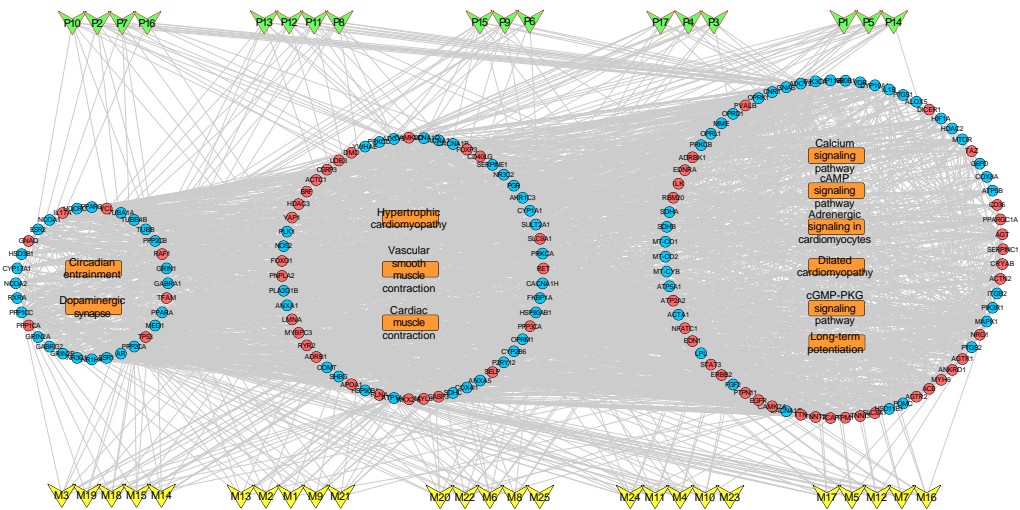

**Figure 3** **VB ingredients-major hubs-pathway network.** Green V-diagrams represent each prototype component in VB; yellow V-diagrams represent each metabolite in VB; round blue nodes represent putative targets of components in VB; round red nodes represent known therapeutic targets for HF; orange rectangles represent top 11 pathways from enrichment analysis of major targets; edges represent interactions among VB ingredients, putative targets, known therapeutic targets for the treatment of VB, and main pathways.

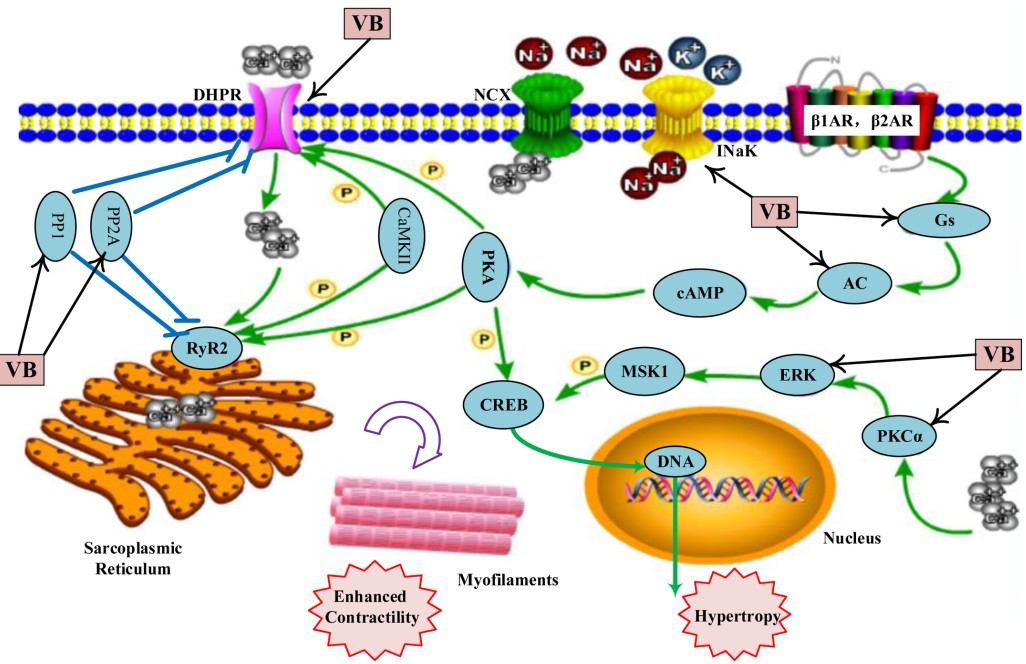

**Figure 4** **Adrenergic signalling in cardio-myocytes influenced by major putative targets of VB components.**

**Table 1   Molecular docking results (top five for each target).**

| Targets | Compound | −CIE (kcal/mol) |
|---|---|---|
| ATP1A1 | Bufotalin | 49.0335 |
| | Cinobufotalin | 46.109 |
| | Cinobufagin | 45.1269 |
| | Cinobufaginol | 44.0494 |
| | $5\beta, 6\alpha$-dihydroxybufalin | 43.3908 |
| GNAS | Cinobufaginol | 55.2668 |
| | $6\alpha$-hydroxybufalin | 53.1182 |
| | 19-oxo-desacetylcinobufagin | 52.2151 |
| | 1,5-dihydroxyldesacetylcinobufagin | 49.9538 |
| | $1,12\beta$-dihydroxycinobufagin | 49.4652 |
| MAPK1 | Cinobufaginol | 55.1432 |
| | Cinobufagin | 54.0654 |
| | Bufotalin | 53.1891 |
| | 1,5-dihydroxyldesacetylcinobufagin | 51.5985 |
| | 12-hydroxyl-cinobufagin | 51.2183 |
| PRKCA | 19-oxo-bufalin | 42.0454 |
| | Marinobufagin | 39.3428 |
| | Hellebrigenin | 38.6171 |
| | $5\beta, 6\alpha$-dihydroxybufalin | 37.9654 |
| | Resibufogenin | 37.8932 |

ATP1A1, GNAS, MAPK1 and PRKCA, which may be the key targets involved in VB for the treatment of HF. Herein, we selected four representative pairs of binding interactions to illustrate how the four targets bound to their corresponding components (Fig. 5). The interplay between ATP1A1 and bufotalin was depicted in Figs. 5A and 5B. The hydroxyl groups on bufotalin could form three hydrogen bonds with SER209, ARG191 and VAL712. Another key residue which involved in interaction was MET157. The binding mode of GNAS and cinobufaginol was depicted in Figs. 5C and 5D. The hydroxyl groups could bind with LEU171, MET255, ASN254 and LYS300 by forming hydrogen bonds. The carbonyl on the lactone ring could also form hydrogen bond with LYS305. In addition, the lactone ring could bind with GLU164 and LYS305 via pi-anion and pi-cation interactions. Other interactions including alkyl and pi-alkyl were connected with ALA303, TYR163 and LEU296. The action mode of MAPK1 and cinobufaginol was depicted in Figs. 5E and 5F. Cinobufaginol could form five hydrogen bonds with LYS112, LYS52 and TYR34. In addition, the lactone ring could bind with ASP109 via pi-anion interaction. Other interactions including alkyl and pi-alkyl were connected with VAL37. The interplay between PRKCA and 19-oxo-bufalin was depicted in Figs. 5G and 5H. The hydroxyl groups of 19-oxo-bufalin could form two hydrogen bonds with PRO202 and LYS230. Another key residue which involved in interaction was LEU200.

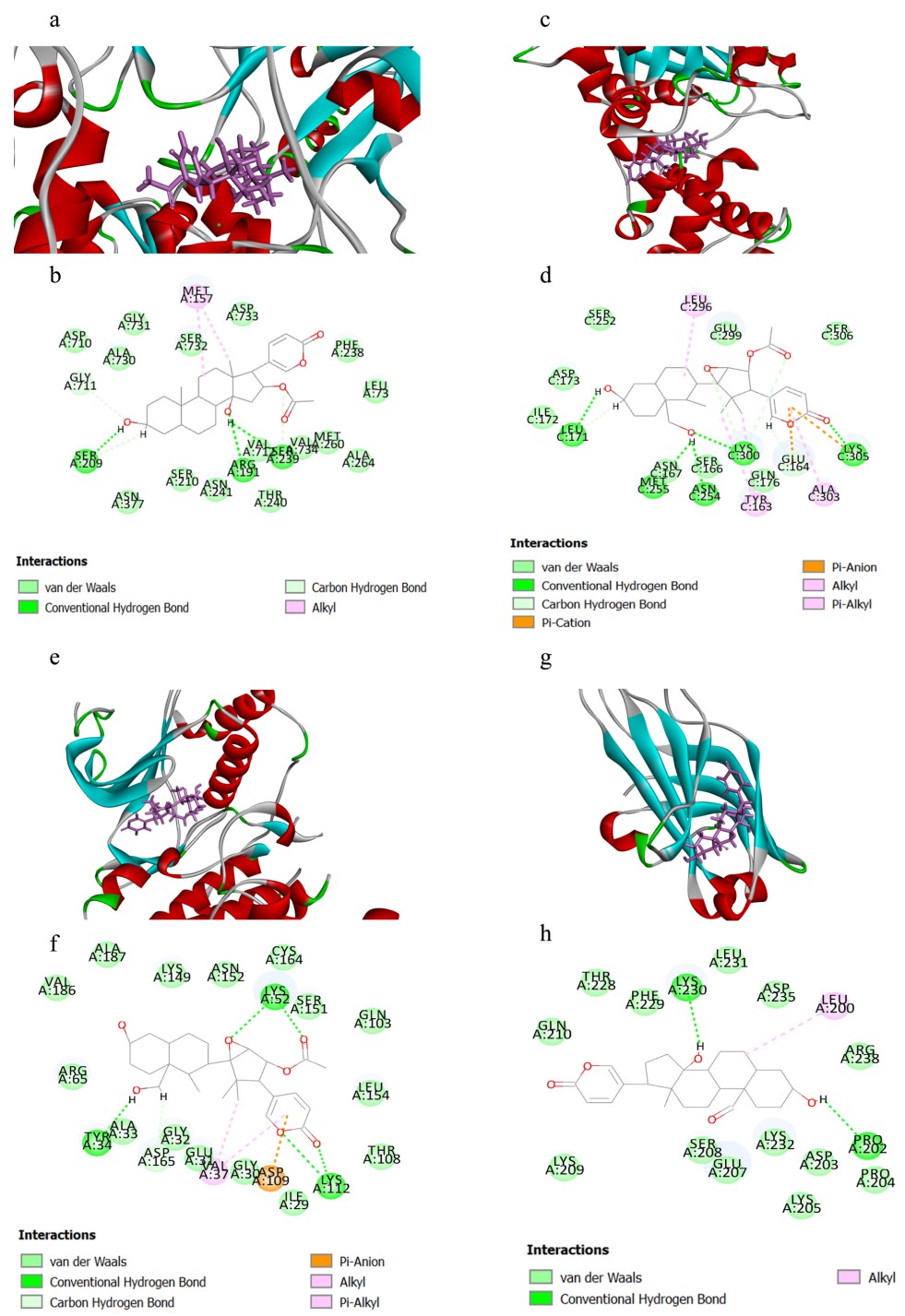

**Figure 5  The binding modes of the selected compounds and targets.** (A) Schematic (3D) representation and (B) Schematic (2D) representation of the interplay between bufotalin and ATP1A1 (PDB IDchimeric 3N23). (C) Schematic (3D) representation and (D) Schematic (2D) representation of the interplay between cinobufaginol and GNAS (PDB IDchimeric 3C14). (E) Schematic (3D) representation and (F) Schematic (2D) representation of the interplay between cinobufaginol and MAPK1 (PDB IDchimeric 3O71). (G) Schematic (3D) representation and (H) Schematic (2D) representation of the interplay between 19-oxo-bufalin and PRKCA (PDB IDchimeric 4DNL). Active site amino acid residues were represented as tubes, while the compounds were shown as a stick model with purple color.

## DISCUSSION

Many pathways are involved in adrenergic signalling for the regulation of cardiac contractile function. Among them, the best described is the mechanism mediated by $\beta$-adrenergic receptor ($\beta$-AR)–Gs–adenylate cyclase (AC) pathway (*Baker, 2014*). Activation of $\beta$-AR–Gs–AC plays an important role in increasing heart rate and force of myocardial contraction (*Chen et al., 2020*; *Santulli & Iaccarino, 2016*). According to our predicted results, VB could regulate $\beta$-AR–Gs–AC pathway by targeting Gs (GNAS) and AC (ADCY2 and ADCY5). The regulation of $Ca^{2+}$ homeostasis is also important for the cardio-myocyte excitation and cardiac electrical activity (*Arakelyan et al., 2007*). According to our predicted results, VB could regulate by targeting PP1 (PPP1CC), PP2A (PPP2CA, PPP2CB), and LTCC (CACNA1C, CACNA1D). $Na^+/K^+$-ATPase, a ubiquitous membrane protein composed of two subunits denoted as $\alpha$ and $\beta$, is also a critical regulator in maintaining the balance of $Ca^{2+}$ in cardio-myocytes (*Orlov et al., 2020*; *Šeflová et al., 2017*). An increasing body of evidence suggests that bufadienolides in VB possess inhibition effects on $Na^+/K^+$-ATPase (*Orlov et al., 2020*; *Sousa et al., 2017*), such as bufalin (*Lan et al., 2018*; *Laursen et al., 2015*), cinobufagin (*Wang, Sun & Heinbockel, 2014*), marinobufagenin (*Strauss et al., 2019*), arenobufagin (*Cruz Jdos & Matsuda, 1993*), and hellebrigenin (*Moreno et al., 2013*). Therefore, the cardiotonic effect of VB may be mainly through the suppression of $Na^+/K^+$-ATPase. Inhibition of the $Ca^{2+}$/PKC $\alpha$/ERK1/2 signal pathway plays a significant role in attenuating the progression of heart failure (*Braz et al., 2004*; *Molkentin & Robbins, 2009*). Several components from VB have been reported to inhibit the activity PKC$\alpha$ and ERK, such as bufalin (*Wu et al., 2015*), marinobufagin (*Bagrov et al., 2000*; *Fedorova et al., 2003*) and cinobufagin (*Baek et al., 2015*).

Based on the data analysis above, the bufadienolides may be the main active components in VB, which exert anti-HF effects via synergistically acting on multiple targets in multiple pathways. Among them, the positive inotropic effect of VB produced through the inhibition of $Na^+/K^+$-ATPase has been demonstrated in many basic researches and clinical practices. The molecular docking results indicated that the representative compounds could connect with the active-site residues via various noncovalent interactions, including the hydrogen bonding, pi-alkyl, pi-anion and pi-cation, etc, which was valuable for understanding of the action mechanisms of VB. In addition, according to -CIE values, bufotalin, cinobufaginol and 19-oxo-bufalin showed the best performance and thus were considered as the potential active leading compounds of the corresponding targets. Further researches on other potential targets or pathways are required to validate the predicted results.

## CONCLUSION

In summary, the active components of VB and their synergistic mechanisms for alleviating HF were successfully unveiled by network pharmacology coupled with molecular docking approach. The adrenergic signaling involved in cardiac muscle contraction process was found to be mainly responsible for the anti-HF effect of VB in silico. Four core targets and their corresponding leading compounds were identified, which may provide valuable information for further experimental validations and drug discovery.

### Funding

This work was supported by the National Traditional Chinese Medicine Clinical Research Base (Grant NO. [2020]33). The funders had no role in study design, data collection and analysis, decision to publish, or preparation of the manuscript.

### Grant Disclosures

The following grant information was disclosed by the authors:
National Traditional Chinese Medicine Clinical Research Base: [2020]33.

### Competing Interests

The authors declare there are no competing interests.

### Author Contributions

- Wei Ren, Zhiqiang Luo and Fulu Pan performed the experiments, authored or reviewed drafts of the paper, and approved the final draft.
- Jiali Liu, Qin Sun, Gang Luo, Raoqiong Wang and Qingrong Pu analyzed the data, prepared figures and/or tables, and approved the final draft.
- Haiyu Zhao and Baolin Bian performed the experiments, authored or reviewed drafts of the paper, provide software, and approved the final draft.
- Xiao Xiao analyzed the data, prepared figures and/or tables, authored or reviewed drafts of the paper, and approved the final draft.
- Sijin Yang and Guohua Yu conceived and designed the experiments, authored or reviewed drafts of the paper, and approved the final draft.

### Data Availability

The raw data are available in the Supplemental Files.

### Supplemental Information

Supplemental information for this article can be found online at http://dx.doi.org/10.7717/peerj.10107#supplemental-information.

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
