# Peer review of "Integrated network pharmacology and molecular docking approaches to reveal the synergistic mechanism of multiple components in Venenum Bufonis for ameliorating heart failure"

_PeerJ, doi:10.7717/peerj.10107_

## Round 0.1 · original submission · Major Revisions

Your manuscript was revised by 3 independent reviewers. They are indicating profound revision, including the necessity of re-analyzing part of the data (and, consequently, re-interpreting the new outcomes). The clarity and the rationale of the selected approaches (for instance, GSEA instead of DAVID; the authors were assuming that all VB-related targets were effective; the paper structure is really complicated, emerging 4 databases (FDA drug-target, drug bank, OMIM, String-db), 4 different bioinformatics tools (MedChem Studio: drug-drug similarity through chemical components; Cytoscape network analyzer; David for enrichment analysis; docking in Discovery Studio), and other efforts such as collecting VB components, etc.), have also to be clearly addressed and corrected. There are also several concerns about data presentation (clarity) and the absence of methodological details. Please, if you are willing to do all the work required, please, indicate clearly in the rebuttal letter where the changes were introduced in the newly revised manuscript. Provide also a scheme of the methodologies used and, most importantly, provide (as supplementary material) details about the steps made in each of the approaches used (I mean provide a list containing all the steps you did in each part of the analysis). Note that before sending the paper for the second round of revision, I will read the new manuscript with detail.

Reviewer 1 ·

Basic reporting

NA

Experimental design

NA

Validity of the findings

NA

Additional comments

In this manuscript “Integrated network pharmacology and molecular docking approaches to reveal the synergistic mechanism of multiple components in Venenum Bufonis for ameliorating heart failure” Ren et al. have performed network and molecular docking analysis of VB and heart failure associated targets and revealed the role of adrenergic signaling. Although authors have done multiple bioinformatics analysis but manuscript in its current form lacks key methodological details and biological insights gained from the current analysis. Following are the comments that need to be addressed to strengthen the manuscript:

Major comments:

1. Section 3.1: What are 260 potential targets of components of VB. Authors should discuss them in detail. Similarly authors should also discuss the heart failure related targets they have identified. Is there any overlap between VB and heart failure related targets?

2. Why enrichment analysis was not done using GSEA instead of DAVID? GSEA is more powerful method to identify enriched pathways and GO terms. What does x axis and percentage implies in Figure2? It is not clear if these are FDR corrected p-values? Use full names of MF, BP and CC instead of abbreviations.

3. Network diagram in Figure 3 is not informative at all. What does three circles represent? Instead of mapping interactions using STRING authors can use other experimentally validated datasets (eg. iRefWeb) to make this figure interpretable. Figure text is unreadable. How many hubs were identified? Please discuss biological role of these hub genes.

4. Authors have not provided the technical details of docking experiment? Is this a rigid or flexible docking method? How many docking runs were carried out? Discuss the biological implications of the results. What are the unit’s of binding energy?


Minor comments:

5. Line 103: What does the confidence score implies to identify VB compound –drug pairs? Is it a drug and Chinese medicine similarity score?

6. Discussion section of the manuscript is verbose and should only focus on the biological insights gained from this analysis and its implication for novel drug usage for heart failure.

7. Figure quality is very poor and text font is unreadable.

Reviewer 2 ·

Basic reporting

Overall speaking, the writing was acceptable.
More details should be added to the docking part.

Experimental design

The paper structure is really complicated, emerging 4 databases (FDA drug-target, drugbank, OMIM, String-db), 4 different bioinformatics tools (MedChem Studio: drug-drug similarity through chemical components; cytoscape network analyzer; David for enrichment analysis; docking in Discovery Studio), and other efforts such as collecting VB components, etc.
Despite these efforts, it looks like the stacking of all these fancy methods.

Validity of the findings

When the authors merged VB-related targets and HF-associated targets and analyzed the network features of the merged dataset. The authors were assuming that all VB-related targets were effective. This means that instead of analyzing how each of VB-related targets might affect heart function, the authors were treating these two sets of genes equally and analyzing the overall function of the merged PPI network. It’s not surprising that a lot of HF-related pathways were extracted out.

Additional comments

Review of Manuscript 49255v1, "Integrated network pharmacology and molecular docking approaches to reveal the synergistic mechanism of multiple components in Venenum Bufonis for ameliorating heart failure "

Venenum Bufonis (VB, or Chan Su in China) has been extensively used as traditional Chinese medicine (TCM) for treating heart failure (HF) since ancient times. Similar to a lot of other TCM, the detailed molecular mechanisms of how VB might work are largely unknown.
In the current paper, the authors applied integrated computational biological methods to address how various components of VB might work. Firstly, 42 components of VB were collected from previous studies. Then, the VB-related targets were inferred by the relationship of VB components----Known FDA drug similarities ---- Drug targets. Meanwhile, the HF-associated targets were inferred from two sources: Drugbank HF targets(109 targets) and OMIM human genetics(199 targets). The linkage between VB-related targets and the HF-associated targets was addressed through PPI network from String-db. The network structure was analyzed using Cytoscape Network Analyzer plugin to find the hubs. 158 hubs, including 93 VB-related and 65 HF-associated targets, were found important based on the network topology. The functions of these hub genes were addressed through gene enrichment analysis using GO/KEGG genesets. Potential pathways were discussed and interpreted as the potential functions of VB. Finally, molecular docking between VB components and their potential targets were performed using the Discovery Studio platform.
The paper structure is really complicated, emerging 4 databases (FDA drug-target, drug bank, OMIM, String-db), 4 different bioinformatics tools (MedChem Studio: drug-drug similarity through chemical components; Cytoscape network analyzer; David for enrichment analysis; docking in Discovery Studio), and other efforts such as collecting VB components, etc. However, despite all these work, the hierarchical nature of the architecture of this paper makes it not suitable to publish at the current stage, and scientific research is not a stacking of what you could do using your currently available data/methods.
When the authors merged VB-related targets and HF-associated targets and analyzed the network features of the merged dataset. The authors were assuming that all VB-related targets were effective. This means that instead of analyzing how each of VB-related targets might affect heart function, the authors were treating these two sets of genes equally and analyzing the overall function of the merged PPI network. It’s not surprising that a lot of HF-related pathways were extracted out.
Other comments:
1. Why not do the gene enrichment analysis of the VB-related targets directly.
2.From line170-180, the author could list them in a table
3.Figure 3, larger fond size,
4.Figure 4, color is too complicated. Because of the nature of PPI network, it’s almost impossible to tell whether it’s activation or inhibition. Still, the authors should try their best to note each pathway whether it’s activation or inhibition.
5. In the docking part, the author should write it clearly how they perform the docking in detail. Such as, what are the targets? Is it all the hub genes or only the VB-related targets? If it was the hub genes, which part of the genes was selected for docking? Regulatory domain? enzyme domain? Or structure domain? What are the small molecules in the docking? Is it the VB-related components? For the CIE, for each of the targets, is there any gold standard of what value makes sense? Or what’s the CIE for a known target-small molecule for each target?

Reviewer 3 ·

Basic reporting

- There contains many grammatical errors and typos in this manuscript. The authors should re-check and revise carefully. The use of language should be improved significantly.
- Quality of figures needs to be improved. Now there are some parts that cannot be seen clearly in the manuscript.

Experimental design

- Methodology hasnot been explained well, it makes difficult for other people in reproducing their results. The authors should explain their methods clear and transparent, not only the use of some packages and software.
- What is the confidence rate of retrieving data from literature?
- GO database and analysis had been used in biomedical works such as PMID: 31921391 and PMID: 31277574. Therefore, the authors should refer more works in this description.
- Why did section 3 is "results and discussion", but section 4 is "discussion" again?
- Why was the confidence score of greater than 0.6 considered as the target proteins of the known drugs?

Validity of the findings

- There are many metrics related to molecular docking analysis. Why did the authors only use "-CIE"?
- The application of this study is too narrow. It is important that the authors discuss and compare their works with some general studies on molecular docking-based or network pharmacology-based Venenum Bufonis.
- The authors should compare their performance results with the previously published works on the same dataset.
- What are orange and green chars in Fig. 2?

Additional comments

No comment

---

## Round 0.2 · Minor Revisions

Thank you for revising your manuscript in accordance with the reviewers' comments. I have looked at the manuscript and I found two points that you have to consider.

In the discussion between lines 292-296, the term notable is too vague and too strong. Either delete it or provide the efficacy of each component (I mean for cited bufadienolides) as IC50 to inhibit the Na+, K+-ATPase.

In the conclusion, please change the sentence appearing at lines 400 and 401 to "The adrenergic signaling involved in cardiac muscle contraction process was found to be mainly responsible for the anti-HF effect of VB in silico." Otherwise, the sentence can give the impression that you have worked either in vitro or in vivo.

Reviewer 3 ·

Basic reporting

No comment

Experimental design

No comment

Validity of the findings

No comment

Additional comments

My previous comments have been addressed satisfactorily.

---

## Round 0.3 · accepted · Accept

Thank you for correcting the manuscript according to my few suggestions.